# Linearithmic Clean-up for Vector-Symbolic Key-Value Memory with Kroneker Rotation Products

**Ruipeng Liu**                                                    RLIU02@SYR.EDU
*Syracuse University*
**Qinru Qiu**                                                      QIQIU@SYR.EDU
*Syracuse University*
**Simon Khan**                                           SIMON.KHAN@US.AF.MIL
*Air Force Research Laboratory*
**Garrett E. Katz**                                             GKATZ01@SYR.EDU
*Syracuse University*

**Editors:** Leilani H. Gilpin, Eleonora Giunchiglia, Pascal Hitzler, and Emile van Krieken

## Abstract

A computational bottleneck in current Vector-Symbolic Architectures (VSAs) is the "clean-up" step, which decodes the noisy vectors retrieved from the architecture. Clean-up typically compares noisy vectors against a "codebook" of prototype vectors, incurring computational complexity that is quadratic or similar. We present a new codebook representation that supports efficient clean-up, based on Kroneker products of rotation-like matrices. The resulting clean-up time complexity is linearithmic, i.e. $\mathcal{O}(N \log N)$, where $N$ is the vector dimension and also the number of vectors in the codebook. Clean-up space complexity is $\mathcal{O}(N)$. Furthermore, the codebook is not stored explicitly in computer memory: It can be represented in $\mathcal{O}(\log N)$ space, and individual vectors in the codebook can be materialized in $\mathcal{O}(N)$ time and space. At the same time, asymptotic memory capacity remains comparable to standard approaches. Computer experiments confirm these results, demonstrating several orders of magnitude more scalability than baseline VSA techniques.

## 1. Introduction

A core issue in neurosymbolic systems is how to embed structured information. Vector-Symbolic Architectures (VSAs) comprise one important embedding approach (Kleyko et al., 2022b; Plate, 2003; Gayler, 1998; Kanerva, 1997). In VSAs, vectors of a fixed dimension $N$ are used to represent individual symbols as well as structured collections of symbols. The latter are represented by composing carefully-crafted vector operators. These well-understood operators can make VSAs more interpretable, and more data- and compute-efficient, than learned embeddings (Roussel, 2023). The vector representation also facilitates integration with transformers and other deep architectures, sometimes reducing their computational complexity as a result (Alam et al., 2023). In many cases, VSA operators can be parallelized on specialized hardware and scaled to very large $N$. In this context, the field of VSAs is also known as hyper-dimensional computing (Kleyko et al., 2022a).

One challenge in VSAs is that accessing structured data is a noisy process. In order to reliably retrieve information stored by a VSA, the noisy vectors that are retrieved must somehow be "cleaned-up." Existing clean-up techniques compare noisy vectors against a "codebook" of prototype vectors (most commonly the individual symbol embeddings) and

have quadratic or similar computational complexity (Kleyko et al., 2022b). Therefore, the clean-up step is a critical bottleneck for current VSAs.

This paper contributes a new form for the codebook that supports efficient clean-up, reducing computational complexity from quadratic to linearithmic, i.e., $\mathcal{O}(N \log N)$. The codebook is a generalized version of a Sylvester Hadamard matrix, whose recursive structure enables the efficient clean-up. We show empirically that this new representation has comparable memory capacity to prior methods, despite the substantial time savings during clean-up. We also show that efficient clean-up facilitates *mutable* VSA memory, since every overwrite operation must reliably retrieve old data in order to erase it properly.

It is worth noting that Alam et al. (2024a) also recently explored an application of Hadamard matrices to VSAs, but for a different purpose. They use Hadamard matrices to derive a novel binding operation, whereas we use a generalized version of Hadamard matrices to derive a novel vector embedding and clean-up method.

## 2. Background

### 2.1. Vector-Symbolic Architectures and Clean-up Methods

VSAs generally involve three types of vector operators:

- **Binding** associates two vectors, analogous to forming a key-value pair.

- **Superposition** forms a set of vectors, analogous to storing multiple key-value pairs in an associative array.

- **Unbinding** retrieves the value associated with a given key.

In all cases, the operators return a new vector representing the result. There are many VSA variants that implement these operators in different ways, using mechanisms such as element-wise multiplication (Gayler, 1998) or element-wise XOR (Kanerva, 1997). VSAs can also represent structures more complex than associative arrays, by including additional vector embeddings of sequence positions or memory pointers.

The vector embeddings for individual symbols comprise a codebook $V$, which can be viewed as a matrix whose $i^{th}$ row vector, denoted $V_i$, is the embedding of the $i^{th}$ symbol. Unbinding operators generally return a noisy vector $u$, which does not exactly match any $V_i$. "Clean-up" finds and returns the closest match to $u$ in $V$. The most basic clean-up approach is a direct computation of dot-product similarity, i.e.:

$$\text{cleanup}(u) = V_{i^*}, \quad \text{where} \quad i^* = \underset{i}{\text{argmax}} \, (Vu)_i. \tag{1}$$

Most VSA methods and analyses use dot-product similarity for clean-up, or closely related metrics such as cosine similarity and Hamming distance for binary vectors (Kleyko et al., 2022b; Yu et al., 2022; Thomas et al., 2021). More sophisticated clean-up methods use various forms of auto-associative memory (Stewart et al., 2011; Steinberg and Sompolinsky, 2022) and handle more complex structure than associative arrays (Frady et al., 2020; Kent et al., 2020). Another approach removes noise by averaging over multiple VSA instances, inspired by Redundant Arrays of Inexpensive Disks (RAID), but the number of instances must be scaled linearly with the number of items to be stored (Danihelka et al., 2016).

To our knowledge, the most efficient clean-up methods to date are based on random linear codes (Raviv, 2024). Their time complexity is either $\mathcal{O}(N \cdot |V|)$ or $\mathcal{O}(N^2 - N \log |V|)$, depending on which algorithm is used, where $|V|$ is the number of vectors in the codebook. Moreover, they do not store $V$ explicitly in computer memory, but only the generator and parity-check matrices for $V$'s linear code, which are much more compact. However, if $|V|$ approaches or exceeds $N$, the time complexity is still essentially quadratic.

## 2.2. Holographic Reduced Representations

Our clean-up method is designed for a specific VSA due to Plate (1995, 2003), known as "holographic reduced representations" (HRRs). In standard HRRs, each vector in the codebook has its entries sampled independently and identically from a distribution with mean 0 and variance $1/N$. Examples of such distributions include the normal distribution $\mathcal{N}(0, 1/N)$, and the discrete uniform distribution over $\pm 1/\sqrt{N}$.

HRRs implement superposition with element-wise addition, while binding and unbinding are circular convolution $\circledast$ and correlation $\#$, respectively:

$$(b \circledast v)_\ell = \sum_{k=0}^{N-1} b_k v_{\ell-k} \qquad (a \# t)_i = \sum_{j=0}^{N-1} a_j t_{i+j}, \tag{2}$$

where subscripts denote vector indices and are taken modulo $N$. For example, the vector $t = (a \circledast u) + (b \circledast v)$ represents an associative array containing two key-value pairs: $(a, u)$ and $(b, v)$. In this example, the unbinding operation $(a \# t)$ will return a noisy version of $u$, i.e., the value previously associated with $a$.

Plate (1995) showed that when codebook vectors are sampled as described above, the HRR memory capacity scales linearly with $N$, and unbinding works correctly in expectation. However, the variance of the unbinding process is also on the order of $1/N$. Therefore, the signal-to-noise ratio is very low and clean-up is essential for reliable retrieval. Plate's original clean-up implementation used direct $\mathcal{O}(N^2)$ computation of dot-product similarity.

The binding and unbinding operations themselves have only $\mathcal{O}(N \log N)$ complexity, since convolution and correlation can be implemented by fast Fourier transform (Heideman et al., 1985). It would therefore be ideal if clean-up is also $\mathcal{O}(N \log N)$. We will show that this is possible using a generalized version of Sylvester Hadamard matrices.

## 2.3. Sylvester's Hadamard Matrix Construction

Hadamard matrices are square, symmetric, orthogonal matrices with binary $\pm 1$ entries. A Hadamard matrix $H^{(k)}$ of shape $2^k \times 2^k$ can be constructed by the following recursive method, given by Sylvester (1867):

$$H^{(0)} = [1] \qquad H^{(k+1)} = \begin{bmatrix} H^{(k)} & H^{(k)} \\ H^{(k)} & -H^{(k)} \end{bmatrix}. \tag{3}$$

This construction can also be expressed as a repeated Kroneker product:

$$H^{(K)} = \bigotimes_{k=1}^{K} \begin{bmatrix} 1 & 1 \\ 1 & -1 \end{bmatrix}. \tag{4}$$

Our core idea is to use matrices like these as codebooks, and leverage their recursive structure to achieve $\mathcal{O}(N \log N)$ clean-up. For standard Sylvester Hadamard matrices, it is known that the fast Walsh-Hadamard transform (Fino and Algazi, 1976) computes the matrix-vector product $Hu$ required for clean-up in $\mathcal{O}(N \log N)$ time. However, standard Sylvester Hadamard matrices do not work well with the HRR operators, since $H_i^{(K)} \circledast H_j^{(K)} = \mathbf{0}$ for many row pairs $i \neq j$, so one cannot use rows as both keys and values. This is partially fixed by using random vectors to embed keys, and rows of $H^{(K)}$ to embed values only, since value embeddings are the ones that must be cleaned up. However, even with this fix, we found that memory capacity was much lower than standard HRRs (Section 4, Fig. 2). Fortunately, a generalization of $H^{(K)}$, introduced next, has comparable memory capacity.

## 3. Methods

### 3.1. Kroneker Rotation Products

We propose a construction similar to Sylvester's but using 2D rotation-like matrices:

$$\tilde{H}^{(0)} = [1] \qquad \tilde{H}^{(k+1)} = \begin{bmatrix} \tilde{H}^{(k)}\cos(\theta_k) & \tilde{H}^{(k)}\sin(\theta_k) \\ \tilde{H}^{(k)}\sin(\theta_k) & -\tilde{H}^{(k)}\cos(\theta_k) \end{bmatrix}, \tag{5}$$

where the $\theta_k$'s with $k \in \{0, \ldots, K-1\}$ are spaced uniformly within $(0, 2\pi)$. These matrices are no longer binary-valued, but they are still orthogonal, symmetric, and possess sufficient structure for linearithmic clean-up. We hypothesize that mixing continuous values into Sylvester's construction better approximates the noise properties of standard HRR embeddings, which is borne out by our experiments. Like Sylvester's original construction, ours can also be expressed as a Kroneker product for $K > 0$:

$$\tilde{H}^{(K)} = \bigotimes_{k=1}^{K} \begin{bmatrix} \cos(\theta_{K-k}) & \sin(\theta_{K-k}) \\ \sin(\theta_{K-k}) & -\cos(\theta_{K-k}) \end{bmatrix}. \tag{6}$$

We will refer to this representation as "Kroneker Rotation Product" or "krop" for short. Technically, each $2 \times 2$ factor is the composition of a rotation and reflection, since the cosine is negated in the second column. After normalizing rows to unit length, Sylvester's construction is in fact a special case of krop where $\theta_k = \pi/4$ for every $k$.

### 3.2. Materializing Rows of $\tilde{H}^{(K)}$

It is possible to reconstruct or sample individual rows of $\tilde{H}^{(K)}$ without storing the entire matrix in memory. Only the $K$ angles $\theta_0, ..., \theta_{K-1}$ must be stored, and $K = \log_2 N$. The $k^{th}$ most significant digit in the binary expansion of $i$ indicates whether $\tilde{H}_i^{(K)}$ came from the first or second row of the $k^{th}$ factor in Equation (6). For example,

$$\tilde{H}_0^{(K)} = [c_{K-1}, s_{K-1}] \otimes [c_{K-2}, s_{K-2}] \otimes ... \otimes [c_1, s_1] \otimes [c_0, s_0], \tag{7}$$

where $c_k$ and $s_k$ abbreviate $\cos(\theta_k)$ and $\sin(\theta_k)$. We can compute products like Equation (7) from right to left, with $K-1$ individual Kroneker products of row vectors (not matrices). The $k^{th}$ product requires $2^{k+1}$ scalar multiplications. Therefore, the entire row is constructed with $\sum_{k=1}^{K-1} 2^{k+1} = 2^{K+1} - 4 = 2N - 4$ operations, i.e., $\mathcal{O}(N)$ time. This process is codified in Alg. 1, where $[\cdot, \cdot]$ is vector concatenation and $>>$ and $\wedge$ are bitwise operators.

---

**Algorithm 1:** Codebook vector reconstruction procedure

---

**Input:** codebook parameters $\theta = [\theta_0 \ldots \theta_{K-1}]$, reconstruct index $i$

$v \leftarrow [1]$

**for** $k \in \{0, ..., K-1\}$ **do**

    $b \leftarrow (i >> k) \wedge 1$

    **if** $b == 0$ **then**

        $v \leftarrow [\cos(\theta_k)v, \quad \sin(\theta_k)v]$

    **else**

        $v \leftarrow [\sin(\theta_k)v, \; -\cos(\theta_k)v]$

    **end**

**end**

**return** $v$

---

### 3.3. `krop` Clean-up

As per Equation (1), the goal of clean-up is to compute $\mathrm{argmax}_i(Vu)_i$, where $u$ is a noisy vector that results from unbinding. We will use $\tilde{H}^{(K)}$ as the codebook $V$, and derive here an efficient algorithm to compute $\tilde{H}^{(K)}u$. This algorithm (Alg. 2 below) is similar to the fast Walsh-Hadamard transform, but modified to incorporate the sines and cosines in $\tilde{H}^{(K)}$.

The algorithm works as follows. Note that here we reason about $\tilde{H}^{(K)}$ mathematically, but do not actually store it in computer memory. Based on Equation (5), we can write

$$\tilde{H}^{(K)}u = \left[ \begin{array}{cc} \tilde{H}^{(K-1)}c_{K-1} & \tilde{H}^{(K-1)}s_{K-1} \\ \tilde{H}^{(K-1)}s_{K-1} & -\tilde{H}^{(K-1)}c_{K-1} \end{array} \right] \left[ \begin{array}{c} \hat{u} \\ \check{u} \end{array} \right] = \left[ \begin{array}{c} \tilde{H}^{(K-1)}(c_{K-1}\hat{u} + s_{K-1}\check{u}) \\ \tilde{H}^{(K-1)}(s_{K-1}\hat{u} - c_{K-1}\check{u}) \end{array} \right], \quad (8)$$

where $\hat{x}$ and $\check{x}$ denote the first and second halves of any vector $x$. Equation (8) decomposes the original problem into two sub-problems where the vector dimensions have been halved. To iterate this decomposition further, we recursively define

$$U_0^{(K)} = u \qquad \begin{array}{ll} U_{2i}^{(k-1)} & = c_{k-1}\hat{U}_i^{(k)} + s_{k-1}\check{U}_i^{(k)} \\ U_{2i+1}^{(k-1)} & = s_{k-1}\hat{U}_i^{(k)} - c_{k-1}\check{U}_i^{(k)} \end{array} . \qquad (9)$$

Each $U^{(k)}$ can be viewed as a list containing $N/2^k$ items, where the $i^{th}$ item $U_i^{(k)}$ is the $2^k$-dimensional vector associated with the $i^{th}$ sub-problem at step $k$.

Expanding the recursion down to $k = 0$, we find

$$\tilde{H}^{(K)}u = \left[ \begin{array}{c} \tilde{H}^{(K-1)}U_0^{(K-1)} \\ \tilde{H}^{(K-1)}U_1^{(K-1)} \end{array} \right] = ... = \left[ \begin{array}{c} \tilde{H}^{(0)}U_0^{(0)} \\ \vdots \\ \tilde{H}^{(0)}U_{N-1}^{(0)} \end{array} \right], \qquad (10)$$

where each $U_i^{(0)}$ is a 1-dimensional vector multiplied by the $1 \times 1$ matrix $\tilde{H}^{(0)} = [1]$. Reinterpreting $U_i^{(0)}$ as a scalar, this means that $(\tilde{H}^{(K)}u)_i = U_i^{(0)}$. Therefore, we do not need to work explicitly with $\tilde{H}^{(K)}$ at all: Instead, we can construct $U^{(0)}$ according to Equation (9), compute $i^* = \mathrm{argmax}_i U_i^{(0)}$, and reconstruct the single row $\tilde{H}_{i^*}^{(K)}$ as per Section 3.2.

---

**Algorithm 2:** `krop` clean-up procedure

---

**Input:** codebook parameters $\theta = [\theta_0 \dots \theta_{K-1}]$, vector $u$

$U_0^{(K)} \leftarrow u$

**for** $k \in \{K, \dots, 1\}$ **do**

    **for** $i \in \{0, \dots, N/2^k - 1\}$ **do**

        $U_{2i}^{(k-1)} \leftarrow c_{k-1}\hat{U}_i^{(k)} + s_{k-1}\check{U}_i^{(k)}$

        $U_{2i+1}^{(k-1)} \leftarrow s_{k-1}\hat{U}_i^{(k)} - c_{k-1}\check{U}_i^{(k)}$

    **end**

**end**

**return** $\underset{i}{\operatorname{argmax}}\, U_i^{(0)}$

---

**Complexity Analysis:** Each $U_i^{(k)}$ is a $2^k$-dimensional vector. Computing $U_{2i}^{(k-1)}$ and $U_{2i+1}^{(k-1)}$ from $U_i^{(k)}$ in Equation (9) therefore requires $3{\cdot}2^k$ arithmetic operations: two element-wise multiplications by $s_k$ or $c_k$ and one element-wise addition or subtraction. Furthermore, $N/2^k$ such vector transformations are performed for each $k$. Consequently, the total number of operations at step $k$ is $3 \cdot 2^k \cdot N/2^k = 3N$. Since there are $\mathcal{O}(N)$ operations for each iteration of Equation (9), and $K = \log_2 N$ iterations, the overall complexity is $\mathcal{O}(N \log N)$. The subsequent argmax is $\mathcal{O}(N)$, so the $\mathcal{O}(N \log N)$ recursion dominates.

### 3.4. Sign-Based Clean-up

As a baseline for comparison, we also consider a simpler clean-up procedure based on the binary $\pm 1/\sqrt{N}$ embeddings proposed by Plate (1995). This baseline clean-up simply rounds a noisy vector $u$ to the nearest binary value vector, i.e.:

$$u \mapsto \operatorname{sign}(u)/\sqrt{N}. \tag{11}$$

The computational complexity of sign-based clean-up is $\mathcal{O}(N)$, so more efficient than `krop` clean-up. However, sign-based clean-up does not have the same guarantees. By construction, `krop` clean-up is guaranteed to correctly reconstruct $\operatorname{argmax}_i(Vu)_i$, where the codebook $V$ contains the $N$ rows of $\tilde{H}^{(K)}$. If we instead sample $N$ embeddings from $\{-1/\sqrt{N}, +1/\sqrt{N}\}^N$, of which there are $2^N$ possibilities, sign-based clean-up is not guaranteed to produce one of the original $N$ embeddings. The $2^N$ possibilities introduce many more opportunities for error, and our empirical results confirm that sign-based clean-up is prone to such errors (Section 4).

### 3.5. Computational Resources

All computer experiments were done on a workstation with 8-core Intel i7 CPU and 32GB of RAM, Fedora 39 Linux, Python 3.11.7, and NumPy 1.26.3. The full set of experiments can be completed in about one day; computation time is dominated by the $\mathcal{O}(N^2)$ direct matrix-vector multiplications used for comparison with `krop`, not by `krop` itself. All experiment code is open-source (MIT license) and freely available online.[1]

---

1. https://github.com/garrettkatz/krop

## 4. Empirical Results

### 4.1. Clean-up Efficiency

We first confirmed that `krop` clean-up is substantially more efficient in practice than direct matrix-vector multiplication. For this experiment we constructed `krop` matrices $\tilde{H}^{(K)}$ with $K$ ranging from 1 to 15. For each $\tilde{H}^{(K)}$ we sampled a noise vector $u$ with i.i.d entries from $\mathcal{N}(0,1)$ and cleaned it up twice, once with direct matrix-vector multiplication, and once with `krop` cleanup. For each $K$, this test was repeated 30 times with different i.i.d samples for $u$. We timed each clean-up method, and also confirmed that they always produced the same result. The running times are shown in Figure 1, which highlights the asymptotically lower complexity of `krop` clean-up. Past $K = 10$, `krop` achieves a considerable speed-up.

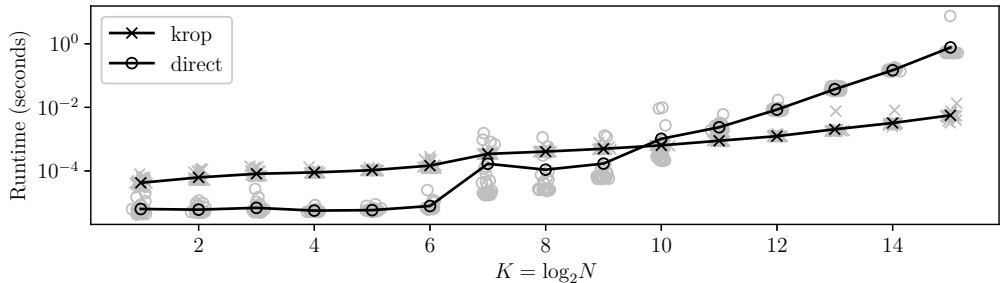

Figure 1: Running times of direct matrix-vector multiply and `krop` cleanup, for individual repetitions (gray) and averages over repetitions (black).

### 4.2. Memory Capacity

We next confirmed that `krop` embeddings exhibit comparable memory capacity to more standard HRR codebooks. Our estimates of memory capacity are based on *retrieval rate*, defined as the fraction of key-value associations that are correctly recalled. Formally, suppose that $M$ key-value pairs $(a^{(m)}, v^{(m)})$ have been stored in a memory trace vector $\mathcal{M} = \sum_{m=1}^{M} a^{(m)} \circledast v^{(m)}$. Then retrieval rate is given by

$$\frac{1}{M} \sum_{m=1}^{M} \mathbb{1}\left[\text{clean-up}(a^{(m)} \oplus \mathcal{M}) = v^{(m)}\right], \tag{12}$$

where $\mathbb{1}[\cdot]$ denotes the indicator function (1 if its operand is True, 0 otherwise). For a given $N$ and $M$, we conduct multiple random trials and calculate retrieval rate in each trial. We say the VSA's "*success* rate" is the fraction of trials in which retrieval rate is 1. For a given $N$, a VSA's "memory capacity" is the largest $M$ for which its *success* rate is 1.

We estimated success rate and memory capacity empirically using 30 independent random trials for each $N$ and $M$. In each trial, the key-value pairs were sampled randomly. Key vectors were always sampled using i.i.d. $\mathcal{N}(0, 1/N)$ entries. Value vectors were sampled uniformly, with replacement, from the codebook. We compared four kinds of codebooks:

- **Normal**: Codebook vector entries sampled i.i.d. from $\mathcal{N}(0, 1/N)$

- **Binary**: Codebook vector entries sampled i.i.d. from $\pm 1/\sqrt{N}$ with equal probability

- **Sylvester**: Codebook vectors are the rows of $H^{(K)}$, normalized to unit length

- `krop`: Codebook vectors are the rows of $\tilde{H}^{(K)}$, with $\theta_k$ uniformly spaced in $(0, 2\pi)$

Normal and binary were cleaned up with direct matrix-vector multiply, while Sylvester and `krop` were cleaned up as per Section 3.3.

We experimentally varied $N = 2^K$ over $2 \leq K \leq 15$, and $M = 2^J$ over $2 \leq J \leq K - 2$. For **normal** and **binary** we capped $K$ at 12 since direct matrix-vector multiplication was less scalable than `krop`. Figure 2 shows success rates for two representative $M$, as well as overall memory capacity (largest $M$ with perfect success rate) for all $N$. Sylvester struggles to retrieve all key-value associations, but `krop` has the same asymptotic capacity as normal and binary, up to a constant factor of 2, while scaling to significantly larger $N$ and $M$. These results corroborate the approximately linear relationship between $N$ and memory capacity derived by Plate (1995), although the asymptotic constants are quite large ($\sim 2^7$).

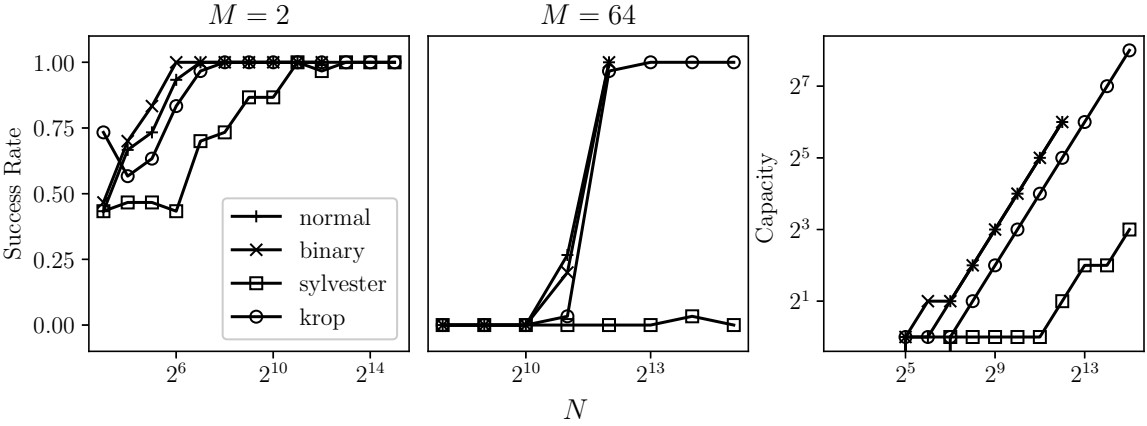

Figure 2: Success rates (left) and memory capacity (right) for each kind of codebook.

### 4.3. Mutable Key-Value Memory

Many real-world tasks require modification of structured data during task execution, meaning that the associations in VSA key-value memory should be easily modified. Efficient clean-up is important for mutable VSA memory, since old data must be reliably retrieved in order to erase it properly. Our last experiment probed this functionality with a series of random trials. Each trial consisted of 30 time-steps, and one association was overwritten in each time-step. As a "ground truth," we maintained a purely symbolic (not VSA) reference memory that saved the most recent value associated with each key. This reference memory was used to compute retrieval rate after every time-step. The idea was that a VSA memory should be *capable* of reliably retrieving any association at any time-step as the task

demands. We compared the following three forms of VSA memory, where $\mathcal{M}_t$ denotes the memory trace at time $t$ and $(a^{(t)}, v^{(t)})$ denotes the new association being written:

- **krop**: This version uses krop embeddings and clean-up. It overwrites an association by unbinding the key's old value, cleaning it up, and then subtracting the old association before adding the new one:

$$v^{\text{old}} \leftarrow \texttt{krop\_cleanup}(a^{(t)} \oslash \mathcal{M}_t^{\texttt{krop}}) \tag{13}$$

$$\mathcal{M}_{t+1}^{\texttt{krop}} \leftarrow \mathcal{M}_t^{\texttt{krop}} - (a^{(t)} \circledast v^{\text{old}}) + (a^{(t)} \circledast v^{(t)}) \tag{14}$$

- **sign**: This version uses $\pm 1/\sqrt{N}$ embeddings and sign-based clean-up:

$$v^{\text{old}} \leftarrow \text{sign}(a^{(t)} \oslash \mathcal{M}_t^{\text{sign}})/\sqrt{N} \tag{15}$$

$$\mathcal{M}_{t+1}^{\text{sign}} \leftarrow \mathcal{M}_t^{\text{sign}} - (a^{(t)} \circledast v^{\text{old}}) + (a^{(t)} \circledast v^{(t)}) \tag{16}$$

- **none**: This version uses $\mathcal{N}(0, 1/N)$ embeddings with no clean-up:

$$v^{\text{old}} \leftarrow a^{(m)} \oslash \mathcal{M}_t^{\text{none}} \tag{17}$$

$$\mathcal{M}_{t+1}^{\text{none}} \leftarrow \mathcal{M}_t^{\text{none}} - (a^{(t)} \circledast v^{\text{old}}) + (a^{(t)} \circledast v^{(t)}) \tag{18}$$

The only candidate for clean-up in this version would be direct matrix-vector multiplication, which would not scale well if required during every overwrite.

Key and value codebooks $A$ and $V$ were sampled/constructed once at the start of the trial, with $|A| = M$ and $|V| = N$. Value codebooks used the embeddings specified above, and key codebooks always used $\mathcal{N}(0, 1/N)$ embeddings. At the start of the trial, each $a \in A$ was associated with its own $v \in V$ sampled uniformly at random. During the trial, $(a^{(t)}, v^{(t)})$ was sampled uniformly at random from $A \times V$.

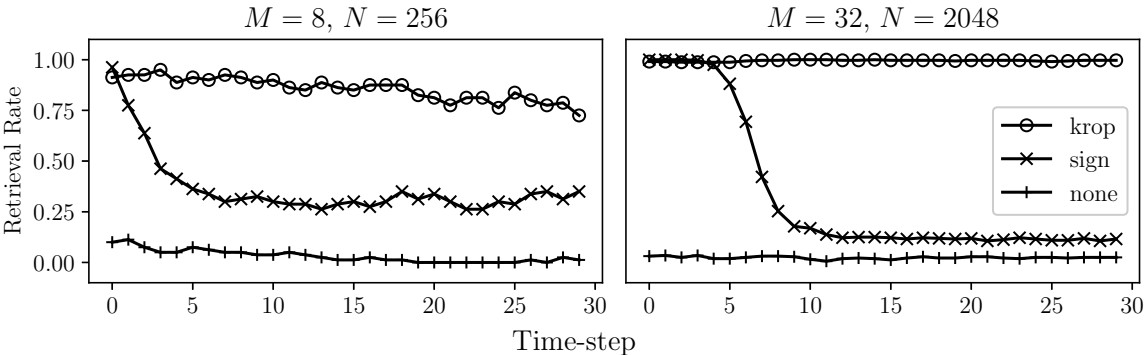

Figure 3: Examples of mutable VSA memory retrieval rates by time-step.

Figure 3 shows per-time-step retrieval rates for two representative $(M, N)$ pairs near the capacity limits observed in Section 4.2. Retrieval rates are averaged over 10 independent trials. krop overwriting is relatively stable over time even when retrieval is imperfect,

whereas sign-based overwriting degrades quickly and overwriting without any clean-up is largely ineffective. More comprehensive results are shown in Figure 4, where retrieval rates are averaged over time-steps and trials. These results confirm that `krop` overwriting is the most scalable and reliable when $N$ is large enough that its memory capacity exceeds $M$.

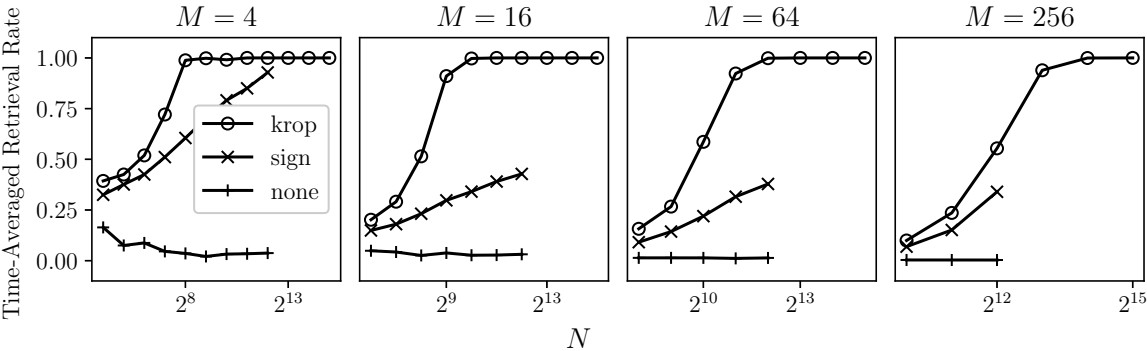

Figure 4: Mutable VSA memory retrieval rates averaged over trials and time-steps.

## 5. Limitations and Future Work

Our present approach uses HRR operators and continuous embeddings. One direction for future work is to extend our methods to discrete embeddings and associated operators, such as MAP (Gayler, 1998), which are better suited to hardware implementation. Future work should also evaluate `krop` on real-world VSA benchmarks, such as extreme multi-label classification (Ganesan et al., 2021) and malware classification (Alam et al., 2024b). One challenge is that real-world data have different distributions than the vectors in the `krop` codebook. This might be addressed by learning linear maps from data vectors to `krop` vectors, or autoencoder maps with bottlenecks if the data dimensionality is too large. Another potential issue is that our current clean-up procedure is not differentiable, hindering its integration with gradient-based optimization. This might be addressed by replacing the argmax with soft attention over all rows of the codebook, again using a Walsh-Hadamard-like transform to efficiently calculate gradients. Lastly, our simple sampling distribution for each $\theta_k$ (uniform over $(0, 2\pi)$) worked for key-value associations chosen uniformly at random, but it remains to be seen whether `krop` is effective for real-world tasks where this distribution could be highly non-uniform. It may be possible to further optimize the choice of $\theta_k$ to achieve codebooks with better noise properties and integration with real-world data.

## Acknowledgments

This research is partially supported by the Air Force Office of Scientific Research (AFOSR), under contract FA9550-24-1-0078. The paper was received and approved for public release by Air Force Research Laboratory (AFRL) on March 7, 2025, case number AFRL-2025-1282. Any opinions, findings, and conclusions or recommendations expressed in this material are those of the authors and do not necessarily reflect the views of AFRL or its contractors.

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
