# OpenReview forum: "Linearithmic Clean-up for Vector-Symbolic Key-Value Memory with Kroneker Rotation Products"
_nesyconf.org/NeSy/2025/Conference — NeSy 2025 Poster_

### Official Review · Reviewer_k8ac · 2025-04-01
**An FFT-like acceleration of matrix-vector products leveraging structured Kroneker product matrices**

**Rating:** 5
**Confidence:** 5

**Review:**

This paper proposes a novel dictionary construction methodology for vector-symbolic architecture, based on Kroneker products of 2D rotation-like matrices. This allows it to efficiently compute similarity computations between a query vector and all elements in a dictionary using a recursive formulation. This reduces the computational search complexity from quadratic to linearithmic.

Overall, the paper is well written and the experiments support most of the claims. However, there are some weaknesses that should be improved:

-	The number of vectors in the codebook is bound by the dimensionality, i.e., $|V| \leq |N|$. Having many more quasi-orthogonal vectors than dimensions is one of the main advantages of VSA. The limitation concerning the number of possible vectors is even acknowledged by the authors in related works when discussing the work (Raviv, 2024). How would this method scale with codebooks where $|V| > |N|$? One obvious approach would be to initialize multiple codebooks with different sampled angles. However, the orthogonality will be lost, and capacity experiments have to prove competitiveness to other VSA approaches that work with randomly drawn, unstructured codebooks.
-	The experimental results should include a comparison to random linear codes (Raviv, 2024). Especially when going beyond $|V| > |N|$.
-	While the overall complexity of the clean-up was reduced from N^2 to NlogN, the proposed clean-up algorithm seems to be quite sequential. This can notably hamper the execution speed on parallel hardware, e.g., GPUs, where most AI workload is deployed nowadays. Hence, speed-ups should be demonstrated on parallel hardware as well.
-	The derivation of the similarity computation (NlogN) seems to be just derived from standard multiplication between a Hadamard matrix and a vector. A similar derivation has already been done in the Fast Fourier Transform (FFT). There exist highly efficient implementations for this kind of workload. Could such techniques also be applied to this workload to parallelize the search?
-	The title suggests an efficient clean-up specifically for key-value memory. However, the method is rather designed for searches in a memory.
-	The paper should discuss recent literature on using Hadamard matrices using VSA [1].

Overall, demonstrations with larger dictionaries ($|V| > |N|$) and comparisons on GPUs should go into a revised version of the paper.

[1] Alam et al., A Walsh Hadamard Derived Linear Vector Symbolic Architecture, NeurIPS 2024.

**Anonymity:**

Remain anonymous

---

### Official Review · Reviewer_6J9d · 2025-04-05
**This paper introduces a novel codebook representation utilizing Kroneker products of rotation-like matrices to address the "clean-up" difficulties in Vector-Symbolic Architectures (VSAs). This exhibits linearithmic time complexity and efficient memory usage, which is a significant improvement over current quadratic approaches. Although the influence from Sylvester's Hadamard matrices is somewhat slow, the mechanism is clear, even though the explanation of the Kronecker Rotation Product could be more straightforward. Although there is a solid theoretical contribution that shows a reduction in time complexity, the limited evaluation on synthetic data and the non-differentiable nature of the clean-up technique raise questions about its practicality, especially when it comes to modern neural network systems. Therefore I rate this paper with a 7.**

**Rating:** 7
**Confidence:** 3

**Review:**

## Methodology

### Kroneker Rotation Products

The paper's main methodological contribution is the introduction of "Kroneker Rotation Products" (krop).  The claim made by the authors that it "better approximates the noise features of standard HRR embeddings" seems a little weak and lacks more thorough explanation. Although the representation is novel, a more elaborate evaluation process could add on the reliability.

### Efficient Clean-up Algorithm

The paper's strongest aspect is the O(NlogN) clean-up algorithm. The detailed description is easy to understand, and the complexity analysis seems to be reliable. However the authors  could elaborate more on how does this linearithmic complexity actually exist in practice, are  there unobserved factors that might decrease the benefits?

### Memory Efficiency

The authors clearly describe how the codebook can be expressed in a compacted manner, and the method's memory efficiency is a plus. Even so, a more thorough examination of the conflict between storage and computation might be beneficial. Does the O(N) time to materialize individual vectors form an issue in memory-constrained applications, or is the O(logN) storage actually beneficial?

## Related Work

### Vector-Symbolic Architectures

The explanation may be a little too brief for people who are unfamiliar with VSAs. Although effective, layering, and unbinding are the three main tasks, their importance and the specifics of the various VSA versions should be further explained.

### Hadamard Matrices

The initial failure of standard Sylvester Hadamard matrices with HRR operators is somewhat skimmed over, but the connection to Hadamard matrices is stated clearly. "Standard Sylvester Hadamard matrices do not work well with the HRR operators," the authors state. A more thorough explanation of this would help the reader see the importance of the statement.

### Clean-up Methods

The evaluation of current cleanup techniques is sufficient, however it lacks the depth of an in-depth assessment.   Although they highlight the disadvantages of quadratic complexity, the authors don't go into detail on the particular flaws in each approach or possible areas for development.

## Experiments & Results

### Clean-up Efficiency

The speed-up is clearly visible in Figure 1, and the tests definitely show that the suggested strategy is more efficient when applied to synthetic data. The sole application of artificial data can restrict the application. Concerns regarding how well these findings translate to real-life scenarios are not sufficiently addressed by the authors. It feels more like a proof-of-concept than a thorough confirmation based on the experiments.

### Memory Capacity

It would have been more beneficial to do a more thorough examination of the memory patterns of usage and any issues that occurred.

### Mutable Key-Value Memory

Although they are a useful addition, the flexible memory experiments have the same issue with synthetic data. Their applicability to actual adaptable memory applications is debatable, despite the fact that they demonstrate the method's ability to handle dynamic data.

## Pros and Cons

### Pros:

* Quality: Although the theoretical advancement is solid, the restricted application only with  synthetic data results in poor experimental reliability.

* Clarity: Although most of the material is understandable, some of the explanations could be clearer and more thorough.

* Originality: The krop representation is a novel contribution.

* Significance: The theoretical significance is clear, but the practical significance can be  questionable due to the evaluation.

* Efficient clean-up with linearithmic time complexity.

* Memory-efficient codebook representation.

* Comparable memory capacity to standard approaches.

* Effectiveness in scenarios involving synthetic flexible memory has been demonstrated.

### Cons:

* Limited Evaluation: There is a significant downside to depending only on data that is artificial.    The framework as a whole can become more reliable by testing on various dataset types.

* Simplistic Distribution: A only slightly justified design choice that raises questions regarding robustness is the choice of a simple distribution for rotation angles.

**Anonymity:**

Remain anonymous

---

### Official Review · Reviewer_7guv · 2025-04-07
**Linearithmic Clean-up for Vector-Symbolic Key-Value Memory with Kroneker Rotation Products**

**Rating:** 6
**Confidence:** 3

**Review:**

Summary
The paper proposes a codebook representation based on generalized Sylvester’s Hadamard Matrix that allows for efficient clean-up in O(N \log N) time, breaking existing known bounds of O~(N^2) for random normal and binary codebooks.  The codebook representation retains comparable capacity with normal codebooks, and experiments show that the cleanup time is significantly lower than normal codebooks for large N values > 1024.

Strengths
The paper proposes a codebook based on extended Sylvester’s Hadamard Matrix that allows for efficient cleanup, which is unexplored in earlier works.
The proposed method also support continuous mutability of the VSA memory with little decrease in rerival rate over time.
The experimental evaluations are quite comprehensive, evaluating the efficiency, capacity, and mutability of the proposed codebook representation.

Weaknesses
The main concern I have is regarding the presentation of the method in section 3, which is lacking in theoretical formality.
Specifically:
All mathematical derivations uses symbolic matrices, while the code presented are directly in Python, which makes it very difficult to correlate the Python code and the discussed theoretical derivation and understand its correctness. It is recommended to rewrite the Python  code into pseudocode that uses the same math notations as in section 3.

The theoretical argument for the core krop cleanup algorithm, it is unclear what the “Derivation” is exactly deriving. Instead, it would be organize this into a concrete theorem and proof. Then, clearly establish the relationship of the theorem with the proposed algorithm.
While there’s a brief statement on the limitation of Sylvester’s Hadamard matrix’s limited capacity, there’s no theoretical argument on the benefits of Kroneker Rotation Products over Sylvester’s Hadamard matrix.

I believe that strengthening the theoretical formality can greatly increase the paper’s contribution to future work using similar ideas.

**Anonymity:**

Remain anonymous

---

### Official Review · Reviewer_omiv · 2025-04-14
**Efficient clean-up in vector symbolic architectures**

**Rating:** 6
**Confidence:** 2

**Review:**

This paper proposed a method for an efficient clean-up method in VSAs which is required for decoding noisy representations. The proposed method is based on a new form of codebook, which is based on Kronecker rotation products (extended version of Sylvester Hadamard matrix) which leads to have the time complexity of this method in the O(nlogn), while the traditional methods are quadratics.

Strengths:
1- It is a well-organized paper with easy-to-follow explanations.
2- Provided code helps to understand the method better.

Concerns:
1- I am not a domain expert in this field, but how would it apply to real-world data? And how can it eventually be integrated into an end-to-end deep learning model?

**Anonymity:**

Remain anonymous